# Conditioning on Dialog Acts Improves Empathy Style Transfer

**Renyi Qu**[1], **Lyle Ungar**[1], **João Sedoc**[2],
[1]University of Pennsylvania, [2]New York University
{requ, ungar}@upenn.edu, jsedoc@stern.nyu.edu

## Abstract

We explore the role of dialog acts in style transfer, specifically empathy style transfer – rewriting a sentence to make it more empathetic without changing its meaning. Specifically, we use two novel few-shot prompting strategies: *target prompting*, which only uses examples of the target style (unlike traditional prompting with source/target pairs); and *dialog-act-conditioned prompting*, which first estimates the dialog act of the source sentence and then makes it more empathetic using few-shot examples of the same dialog act. Our study yields two key findings: (1) Target prompting typically improves empathy more effectively than pairwise prompting, while maintaining the same level of semantic similarity; (2) Dialog acts matter. Dialog-act-conditioned prompting enhances empathy while preserving both semantics and the dialog-act type. Different dialog acts benefit differently from different prompting methods, highlighting the need for further investigation of the role of dialog acts in style transfer.

## 1 Introduction

Expressing empathy is important for communication in applications ranging from customer service to mental health. Although there is no universal definition of empathy, most definitions share a core of having the ability to understand and experience the feelings of others (Cuff et al., 2016).[1] Recent research has explored empathetic response generation, where the language model generates a response that conveys both the intended message and a sense of empathy given the input context (Majumder et al., 2020; Liu et al., 2022; Zheng et al., 2021; Sabour et al., 2022). However, empathetic response generation is not applicable to more sensitive cases such as medicine and therapy, where the content of the responses requires the supervision

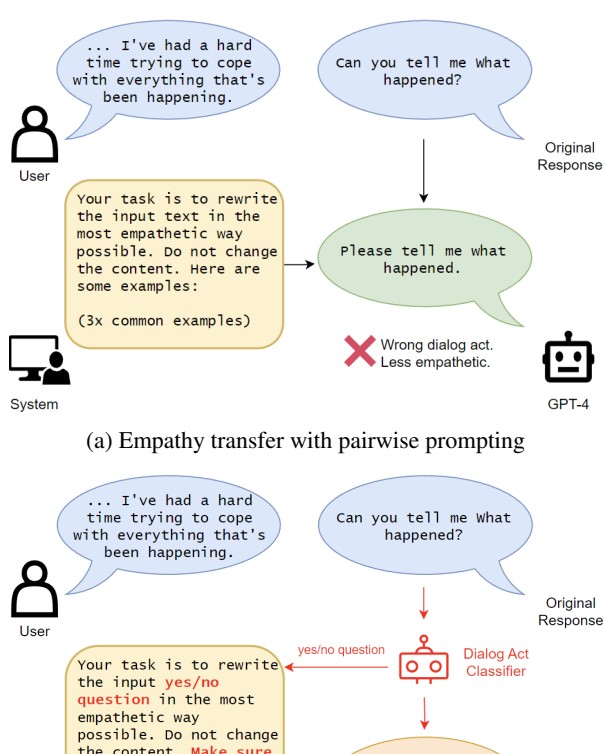

(a) Empathy transfer with pairwise prompting

(b) Empathy transfer with DA-target prompting (explicit)

Figure 1: Example of conditioning few-shot examples and prompt format on the source dialog act. (a) illustrates pairwise prompting, which uses a common set of few-shot example pairs for all inputs; (b) illustrates DA-target prompting (explicit), which identifies the source dialog act and changes the prompt accordingly, as shown in red, resulting in a better performance in both empathy transfer strength and content preservation.

of medical experts. A key issue for state-of-the-art conversational models is hallucination (Dziri et al., 2022; Azamfirei et al., 2023), where the models provide fake or incorrect information in their re-

---

[1]Of course, empathic chatbots do not actually experience feelings; they merely speak as if they did.

sponses. Physicians often (reasonably) do not trust the content of computer-generated responses but use large language models (LLM) such as GPT-4 (OpenAI, 2023) to rephrase their responses to express more empathy.

This is style transfer: paraphrasing a sentence from one style to another without changing its content (Table 1). The input sentence is typically named the source sentence or "prompt", and the output sentence is typically named the target sentence. Style is, unfortunately, often entangled with content: being polite, rude, or angry is hard to separate from e.g. the substance of a product review. Many researchers have worked on this task, but it remains challenging because of many unsolved issues, including the lack of parallel data and the lack of reliable evaluation metrics for different styles (Toshevska and Gievska, 2021; Hu et al., 2022; Jin et al., 2022; Lample et al., 2019).

Few-shot prompting, also called "in-context learning", has proven remarkably good at style transfer, giving performance competitive with more data-intensive fine-tuning (Reif et al., 2022). Despite having some limitations (Webson and Pavlick, 2022), prompting is attractive in eliminating the need for extensive curated data. Previous work compares zero-shot and pairwise few-shot prompting (where there are, for example, content-matched pairs of high and low empathy responses) on text style transfer (Reif et al., 2022). Zero-shot prompting is prone to generating illogical or nonsensical outputs, while pairwise few-shot prompting requires optimal choices of sampled sentence pairs to maximize its performance (Suzgun et al., 2022).

In this paper, we investigate empathy style transfer using prompt-based inference with GPT-4. We propose two sets of novel prompting approaches: *target prompting* and *dialog-act-conditioned prompting*. Target prompting is inspired by the difficulty of collecting counterparts of empathetic sentences, and uses sentences of only the target style as its examples (i.e., highly empathetic sentences). It contrasts with the more common *pairwise prompting*, where the prompts contain semantically matched pairs of high and low empathy responses. Dialog-act-conditioned prompting is inspired by the fact that dialog acts play a crucial role in pragmatics and convey essential information about our cognitive processes, which are closely related to our empathy expressions (Bunt, 2011). We propose two types of dialog-act-conditioned prompting: (a) selecting few-shot examples based on the dialog act of the source sentence and (b) explicitly specifying the dialog act in the prompt format in addition to (a). We test our prompting strategies and focus our evaluations on empathy transfer strength and content preservation. In summary, we find that:

1. Target prompting generates higher empathy than pairwise prompting while preserving a similar level of content

2. Dialog acts matter: conditioning the prompt on them when doing style transfer increases empathy while better preserving source dialog acts. Different dialog acts benefit differently from different prompting methods

## 2 Related Work

**Empathetic Response Generation**   Many researchers developed empathetic dialogue systems as this may improve the overall usefulness of AI systems (Czerwinski et al., 2021). These generally rely on datasets of empathetic dialogs, with early datasets (Rashkin et al., 2019) being followed by large-scale (1M empathetic dialogues) (Welivita et al., 2021) and multi-level datasets of conversations (Omitaomu et al., 2022). Recently, Welivita et al. (2023) created an empathetic response dataset for distress-related topics with comprehensive evaluation metrics. The availability of open-source datasets allowed the creation of many transformer-based empathetic response generation models (Li et al., 2020b; Majumder et al., 2020; Liu et al., 2022; Zheng et al., 2021; Sabour et al., 2022), all of which directly generate empathetic responses from the input prompt. Qian et al. (2023) discovered that a two-stage system (response generation and style transfer) can yield better performance than one-stage models. However, a two-stage system requires two language models to be separately trained on different datasets. More recently, Lee et al. (2022) confirmed that LLMs are capable of generating empathetic sentences through zero-shot prompting, reducing the burden of training and shifting the focus to prompt engineering.

**Text Style Transfer**   One conventional way to do text style transfer was style-content disentanglement, where encoder-decoder structures such as Variational Autoencoders (Bao et al., 2019; Liu

| Entity | Empathy | Content |
|---|---|---|
| input sentence | 2.74 | Yes I'm worried that things like that will continue to happen in the US also. |
| zero-shot response | 1.97 | I understand that such events will persist in the US as well. |
| few-shot target prompting response | 2.84 | I share your concern that such events may persist in the US, causing distress and unease among its people. |
| few-shot DA-target prompting response | 3.23 | I can genuinely understand the concern that events like these may persist in the US, causing a sense of unease and apprehension in our hearts. |

Table 1: Example of input and responses generated by GPT-4 in three different conditions. The computed empathy scores show that the responses have higher empathy when using target prompting. The exact methods are described later in the paper, but "DA-target prompting" uses the dialog act of the prompt.

et al., 2020; Yi et al., 2021) and transformers (Sudhakar et al., 2019; Dai et al., 2019; Lee, 2020) were used to alternate the style embeddings in order to do style transfer. Following the rapid development of transformer models, many researchers conducted experiments on more specific styles such as sentiment (Wu et al., 2019; Pant et al., 2020), formality (Yao and Yu, 2021; Lai et al., 2021), and politeness (Madaan et al., 2020). The most recent work on empathy style transfer is Zhang et al. (2022), where they used an empathy transformer and a discriminator to tune the empathy level. However, all the methods above require data availability and training, making them less efficient.

More recently, prompt-based inference with LLMs was significantly more efficient than fine-tuning, and it eliminated the issue of lack of data availability in text style transfer. Reif et al. (2022) used the prevalent zero-shot and few-shot prompting with GPT-3 (Brown et al., 2020) for arbitrary style transfer. Suzgun et al. (2022) prompts the large language model multiple times to generate multiple responses and reranks them based on three evaluation metrics: transfer strength, semantic similarity, and fluency. However, to our knowledge, none of the text style transfer work has considered dialog acts so far.

**Dialog Act** Dialog act is an indicator of the intent or purpose of a speaker's utterance within a conversation. It has long been acknowledged that understanding dialog acts is crucial for discourse analysis, especially for pragmatic aspects like sentiments and emotions. Cerisara et al. (2018) first attempted to model the joint distribution of dialog acts and sentiments, and Li et al. (2020a) further enhanced the model and acknowledged the strong correlation

between dialog acts and sentiments. Later, Saha et al. (2020) incorporated emotions into dialog act classification and confirmed that emotions affect our dialog act choices; and Bothe et al. (2020) was able to build neural annotators to jointly annotate emotions and dialog acts. More recently, Li et al. (2022) found out that dialog act prediction significantly improved depression detection, hinting at the strong potential of dialog acts in understanding the emotional and mental aspects of discourse.

Dialog acts are rarely, if ever, used in text style transfer, but their use has enhanced performance in other natural language generation tasks (Wang et al., 2020; Peng et al., 2020; Gella et al., 2022). Motivated by these papers, we believe that using dialog acts will benefit empathy style transfer.

## 3 Methods

### 3.1 Problem Setting

Prior work on empathy–and most other–style transfer has considered the source and target styles as two different styles in a binary setting, focusing on the two opposite ends of the style scale, while ignoring the middle area. In contrast, we assume that source and target styles are of the same type but at different positions on a continuous, real-valued scale. Discrete style labels miss the continuous nature of the empathy style (Cuff et al., 2016; Lahnala et al., 2022). Hence, we formulate our empathy transfer task as follows. Given a target style $s$ and a sentence $x$ with a degree of style polarity $p$, empathy style transfer outputs a sentence $y$ with the highest attainable polarity $p'$ of the same style while adequately preserving the sentence semantics. In this paper, we are transferring from lower to higher empathy.

## 3.2 Prompting Strategies

Table 2 summarizes the few-shot prompting methods used in this paper. For more details on the prompt formats, please refer to Table 6 in the Appendix. We are interested in two branches of comparison: *target vs pairwise prompting* and *dialog-act-conditioned vs direct prompting*. The first branch differs in the structure of few-shot examples, while the second branch differs in the conditionality of few-shot examples on the dialog act of the source sentence.

**Target prompting** Inspired by the challenges in pairwise prompting, we use only the target sentences as the few-shot examples without any pairwise comparison. Since non-parallel data is abundant, this method eliminates the effort of searching for or manually creating high-quality counterexamples for style transfer.

**DA-conditioned prompting** We expand pairwise and target strategies to a two-stage process by conditioning the prompt on dialog acts. First, we identify the dialog acts of the source sentence using a dialog act classifier. Then, we map the identified dialog act to its corresponding few-shot examples. Since the examples are of the same dialog act, we expect the model to perform empathy transfer better in terms of naturalness, coherence, and content preservation of the source sentence. Because we do not tell the LLM what the source dialog act is, this approach is considered "implicit".

As an extension to the implicit approach, we explicitly specify the source dialog act in the prompt (example shown in Figure 1). Specifically, we change the identifier words in the original prompt into the dialog act name, and we add one extra sentence to emphasize the preservation of dialog acts. This approach is considered "explicit".

## 3.3 Dialog act categorization

We present the categorization for dialog acts in Table 3, similar to Jurafsky et al. (1997) with some modifications. Rhetorical questions (*qh*) are excluded from the Question category because they function differently from normal questions in a more similar way to other forward expressions. Not all dialog acts are included in the EC dataset, so we excluded the nonexistent ones from the table.

| Category | Dialog Act |
|---|---|
| **Forward Communication** | |
| Statement | sd, sv |
| Question | qo, qr, qw, qy |
| Other Forward | fa, fc, fp, ft, fx, qh |
| **Backward Communication** | |
| Agreement | aa, ad |
| Understanding | a, b, ba, bf, bh, br, by, ^2 |
| Answer | na, ng, nn, no |
| **Other** | |
| Other | ^q, h, +, %, t3, 'none' (unclassified) |

Table 3: Categorization of dialog acts in order to reduce complexity in our analyses. These categories are mostly the same as Jurafsky et al. (1997). Only the dialog acts involved in the EC dataset are included. Acronyms are defined in Appendix Table 5.

# 4 Experiments

## 4.1 Data

We evaluate our prompting methods on the Empathic Conversations (EC) dataset (Omitaomu et al., 2022), which consists of 500 conversations (11,778 utterances) between Amazon MTurkers on 100 articles on various tragic events. Each article contributes a total of 5 conversations. Each conversation consists of 20-25 utterances between 2 MTurkers with its topic strictly focused on the assigned article. This data mimics our daily interactions with empathetic scenarios extremely well and is optimal for our empathy transfer experiment.

Before the experiment, we use a pretrained RoBERTa (Liu et al., 2019) evaluator to calculate the empathy scores of all utterances in a continuous range of 0-4, and we use a pretrained RoBERTa multilabel classifier[2] to predict the dialog acts of all utterances from the set of 42 labels from Switchboard (Jurafsky et al., 1997). When an utterance is classified with multiple dialog acts, we select the one with the highest probability. When an utterance is classified with no dialog act due to ambiguity, we put it into the *none* category.

## 4.2 Conditionality of Empathy on Dialog Acts

Our exploratory analysis of the EC dataset indicates a strong correlation between empathy strength and dialog acts (Figure 4) and demonstrates the following:

- Opinions, Facts, and Agreements have near-

---

[2]Please refer to (Omitaomu et al., 2022) for the performance of the classifiers.

| Prompting Method | Description |
|---|---|
| Pairwise | Each example consists of one unempathetic-empathetic sentence pair of high semantic similarity. |
| DA-Pairwise (implicit) | Pairwise prompting where each example shares the source dialog act. |
| DA-Pairwise (explicit) | Pairwise prompting where each example shares the source dialog act and the prompt explicitly specifies the source dialog act. |
| Target | Each example consists of one empathetic sentence. |
| DA-Target (implicit) | Target prompting where each example shares the source dialog act. |
| DA-Target (explicit) | Target prompting where each example shares the source dialog act and the prompt explicitly specifies the source dialog act. |

Table 2: Few-shot prompting methods. Pairwise prompting is the conventional prompting method used in text style transfer, while the other 5 are novel variations.

Normal empathy score distributions[3]

- Wh-Questions, Yes/No Questions, and Open Questions have empathy scores skewed to the lower end, while Rhetorical Questions have empathy scores centered in the middle

- Forward Closings have empathy scores heavily skewed to the lower end, while Backward Sympathy expressions have empathy scores heavily skewed to the higher end

In addition, low empathy sentences are mainly Forward Closing sentences (*fc*), while high empathy sentences are mainly Opinions (*sv*), Facts (*sd*), and Agreements (*aa*) (Figure 2). Different dialog acts have different empathy score distributions (Appendix B).

### 4.3 Sample Selection

**Test sample selection** If we conduct our experiment only on the utterances with the lowest empathy scores, the results are largely biased towards Forward Closing sentences (Figure 2). To mitigate this bias, we sort all conversations based on the mean empathy score over all utterances within each conversation. We then select the 41 conversations (1016 utterances) with the lowest mean empathy scores. This is beneficial for two reasons. First, each conversation is long enough to include a reasonable variety of dialog acts, thus mitigating sample biases towards any specific dialog act. Secondly, we still obtain a reasonable set of utterances with low empathy scores, which is more suitable for evaluating empathy transfer strength compared

[3]For simplicity, we refer to Opinion Statements as Opinions and Non-opinion Statements as Facts.

to highly empathetic samples. Appendix Figure 3 shows the empathy score distribution and dialog act histogram for the selected test samples.

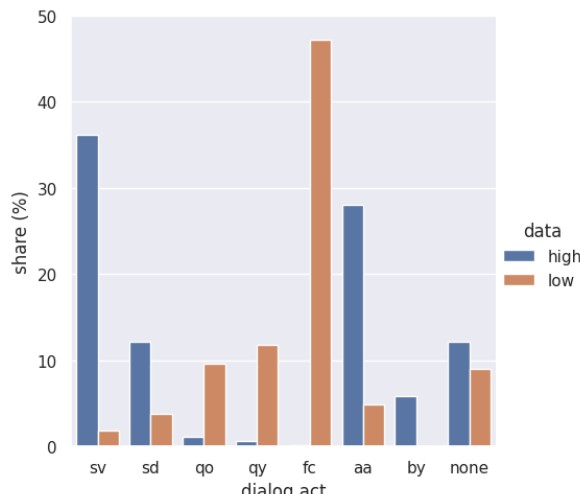

Figure 2: Shares of most frequent dialog acts in the 2000 utterances of the highest empathy (blue) and the lowest empathy (orange) in the EC dataset. Opinions (*sv*), Facts (*sd*), Agreements (*aa*), and Backward Sympathy (*by*) appear much more often in highly empathetic sentences, while Questions (*qo*, *qy*) and Forward Closings (*fc*) appear much more often in lowly empathetic sentences, indicating the dependency of empathy on dialog acts.

**Few-shot example selection** We sample 3-shot examples from GPT-4 by prompting it to generate high and low empathy sentences. A sentence with an empathy score higher than 3 (75% of total score) is accepted as a highly empathetic example, and a sentence with an empathy score lower than 1 (25% of total score) is accepted as a low empathy example.

For Target prompting, we directly use the quali-

fied highly empathetic examples as the few-shot examples. For Pairwise prompting, we take the same highly empathetic examples from Target prompting and ask GPT-4 to rewrite them into an unempathetic version. We repeat this until the rewritten version satisfies the score constraint. Then, we pair the original empathetic sentences and the rewritten unempathetic sentences together as our few-shot examples.

For direct prompting, we use the pretrained RoBERTa classifier to ensure that the selected few-shot examples have different dialog acts from each other to avoid possible conflict with the examples used in DA-conditioned prompting. We use the same qualified examples across all input dialog acts. For DA-conditioned prompting, we explicitly instruct GPT-4 to generate sentences of the selected dialog act and validate their dialog acts using the classifier.

## 4.4 Model

We use GPT-4 (OpenAI, 2023) to perform empathy style transfer. Each test sample is combined with the predefined prompt formats shown in Figure 1 as the full prompt, and GPT-4 generates a transferred sentence with a temperature of 0 and a maximum length of 32. In addition to the few-shot prompting methods in Table 2, we use zero-shot prompting as a baseline, where we directly ask GPT-4 to rewrite the source sentence with no example.

## 4.5 Evaluation

We follow the three categories of evaluation criteria specified in Hu et al. (2022): Transfer Strength, Content Preservation, and Fluency. We also introduce a new metric to monitor GPT-4's tendency to add tokens during style transfer. All of the following evaluation metrics are automatic.

**Transfer Strength**   Transfer strength directly shows how good the style transfer method is. We use a pretrained RoBERTa evaluator to calculate the empathy scores of the generated sentences and compare them with the source sentences. We use the following metrics to assess transfer strength:

- $\Delta$**Emp**: We calculate the empathy score difference within each pair of generated and source sentences in the range $[-4, 4]$, and we report the average over all test samples.

- **Acc**: We calculate the share of samples with increased empathy scores after style transfer

as accuracy. The basic idea of empathy style transfer is to make a sentence more empathetic, so a decrease in empathy score is considered a failure.

**Content Preservation**   While transfer strength is crucial, we do not want to overly change the content of the source sentence. We use the following metrics to assess content preservation:

- **BERTScore**: We obtain the F1 score from BERTScore (Zhang et al., 2019) for each pair of generated and source sentences, and we report the average over all test samples. BERTScore calculates text similarity token by token based on contextual embeddings and outperforms traditional semantic similarity metrics. We use *deberta-xlarge-mnli* (He et al., 2020) as the best-performing variation.

- **BLEURT**: We calculate BLEURT for each pair of generated and source sentences in the range $[0, 1]$, and we report the average over all test samples. BLEU scores (Papineni et al., 2002) are notorious for not resembling human judgments on semantic similarity (Callison-Burch et al., 2006), so we use BLEURT-20, a BERT-based BLEU variation trained on many synthetic examples (Sellam et al., 2020), to get similarity scores closer to human judgments.

- **DA-Acc**: We calculate the share of samples with dialog act unchanged after style transfer as the accuracy for dialog act preservation.

**Other attributes**   We use the following metrics for additional evaluation:

- $\Delta$**Length**: We calculate the difference in the number of words for each pair of generated and source sentences and report the average over all test samples. During the experiment, we noticed that GPT-4 tends to add many tokens to reach the desired outcome for style transfer, so we included this metric to monitor such behavior.

- **Perplexity**: We use GPT-2-Large (Radford et al., 2019) to measure the perplexity of each generated sentence, and we report the average over all test samples.

| Method | Transfer Strength | | Content Preservation | | | Others | |
|---|---|---|---|---|---|---|---|
| | ΔEmp ↑ | Acc (%) ↑ | BERTScore ↑ | BLEURT ↑ | DA-Acc (%) ↑ | ΔLength ↓ | PPL ↓ |
| Zero-shot | 0.23 | 75.98 | **0.72** | **0.63** | 40.78 | **5.24** | 69 |
| Pairwise | 0.59 | 90.85 | 0.68 | 0.59 | 49.80 | 12.34 | 30 |
| DA-Pairwise (implicit) | 0.65 | 93.90 | 0.68 | 0.58 | 54.81 | 11.71 | 33 |
| DA-Pairwise (explicit) | 0.70 | 92.81 | 0.68 | 0.59 | 55.56 | 12.08 | 34 |
| Target | 0.67 | 93.41 | 0.67 | 0.58 | 46.46 | 15.18 | 29 |
| DA-Target (implicit) | **0.80** | **96.56** | 0.68 | 0.58 | 53.05 | 13.04 | 31 |
| DA-Target (explicit) | 0.77 | 95.96 | 0.69 | 0.59 | **56.50** | 11.23 | **28** |

Table 4: Evaluation results. The bolded items are the best values for each metric. The arrows show the direction of better performance of the corresponding metric. Row 1 is the baseline; Row 2-4 are pairwise prompting methods; Row 5-7 are target prompting methods. Target methods transfer better empathy than pairwise methods, and DA-conditioned methods transfer better empathy and preserve dialog acts better than direct methods. No visible difference in semantic similarity is observed among the few-shot prompting methods.

## 5 Results

**Target vs Pairwise** As shown in Table 4, regardless of whether one conditions the prompt on the source dialog act, target prompting always outperforms pairwise prompting in terms of both empathy score and accuracy. This difference is statistically significantly higher for both direct and DA-conditioned methods. Without conditioning on dialog act, sentences generated by target prompting are slightly less similar to the source sentences compared to pairwise prompting, but the difference is nearly negligible. With conditioning on dialog act, the difference in semantic similarity disappears. This is expected since we always explicitly instruct GPT-4 not to change the content (Figure 1). This result suggests that it is unnecessary to collect pairwise samples when doing empathy style transfer through prompting.

**DA-conditioned vs Direct** Regardless of whether we use target or pairwise prompting, conditioning the prompt on the dialog act of the source sentences improved empathy transfer strength while maintaining a similar level of semantic similarity. The difference in empathy transfer strength is statistically significantly higher for both target and pairwise methods. Also, with DA-conditioned prompting, more samples have their dialog acts unchanged after empathy style transfer. However, the difference is not as large as might be expected, as shown by DA-Acc. This could potentially result from the failure modes of the dialog act classifier, as discussed in Section 4.1 and further addressed below in Section 6.

**Implicit vs Explicit** While DA-conditioned methods outperformed direct methods in general, explicitly specifying the source dialog act in the prompt does not necessarily improve the performance of DA-conditioned prompting. For DA-Pairwise prompting, explicit specification transfers higher empathy on average, but implicit prompting achieves higher accuracy, meaning that more samples are successfully converted into their more empathetic versions. For DA-Target prompting, implicit prompting outperforms explicit specification in both the amount of transferred empathy and the success rate of empathy style transfer. This might be a result of how the prompt is formatted. In the explicit version, we inject tokens related to the source dialog act into the prompt, which inevitably diverts some attention weight from the empathy style transfer instruction to the dialog act preservation instruction. As a result, explicit specification leads to a higher accuracy for dialog act preservation than the implicit version. For this case, neither target nor pairwise are statistically significantly different.

**Zero-shot vs Few-shot** Although Reif et al. (2022) showed that few-shot prompting generally performs better than zero-shot on GPT-3, the stunning zero-shot capabilities of GPT-4 on various generation tasks made people question whether few-shot prompting is still needed (OpenAI, 2023; Peng et al., 2023; Bubeck et al., 2023). We find that zero-shot prompting with GPT-4 performs much worse than few-shot prompting. All few-shot methods are better at empathy transfer. All few-shot methods are more likely to preserve the source dialog act after style transfer than zero-shot prompting was. Zero-shot prompting produces highest text similarity scores according to BERTScore and BLEURT, but there are two reasons why this does

not necessarily matter. First, automatic similarity measures are not reliable for measuring content preservation (Shimorina, 2018). Human evaluations are necessary to validate this. Second, if zero-shot prompting does preserve content better than few-shot prompting, the significant lack of transfer strength is a direct drawback for using zero-shot prompting. This is also shown by the difference in sentence lengths. Zero-shot prompting tends to add much fewer words than few-shot prompting does, suggesting that zero-shot prompting does not change the sentences by much, thus yielding higher similarity. The lower similarity scores of few-shot methods may also be caused by the lower similarity between the few-shot examples and the input source sentence since dialog act was the only controlled sentence attribute.

## 5.1 DA-specific results

The performance of different prompting methods varies across the source dialog acts. We cluster the 31 dialog acts in the EC dataset into higher-level categories based on Switchboard (Jurafsky et al., 1997) (Table 3). There are two main categories: forward communication and backward communication, each including three subcategories. The "Statement" category is further separated into "Non-opinion" and "Opinion" during analysis since they comprise most of the sentences involved in the test samples.

We present the results of all seven prompting methods for each dialog act category in Figures 5, 6, and 7 in Appendix E, and describe the major findings here. We conclude that different methods work differently on different dialog act categories.

We find the following DA-specific evaluation results for ΔEmp, style transfer accuracy, and BLEURT (Figure 5):

- DA-Target prompting generally transfers better empathy at a higher accuracy compared to other methods, with the explicit version performing particularly well for facts and the implicit version for others

- Direct target prompting performs better than all pairwise methods for most forward expressions except Facts

- Relative to other categories, empathy style transfer does not boost the empathy strength of Opinions and Questions

- Relative to other categories, the BLEURT scores for Other forward expressions (including forward openings and closings) are particularly low, indicating that it may not be a good idea to do empathy style transfer on such dialog acts due to changes in content

Changes in dialog acts after empathy style transfer for forward communication and backward communication are shown in Appendix Figures 6 and 7.

For forward communications, facts are prone to be converted into opinions, and this tendency is surprisingly much more visible in explicit versions of DA-conditioned prompting. Hence, it seems unwise to tell GPT-4 the source dialog act is a fact when doing empathy style transfer, and it may not be a good idea to use empathy style transfer on facts in the first place. On the other hand, DA-conditioned methods work particularly well with maintaining the dialog acts of Questions and Other forward expressions, but direct methods are slightly better at keeping Opinions as opinions.

For backward communications, the performance is vastly different. DA-conditioned prompting works very well with Agreements, and explicit specifications of Agreements help preserve this dialog act. Performance on Answers and Other backward expressions is problematic, but note that the sample sizes of Answers and Other backward expressions are small, so the results may not be representative of the overall samples. However, these results may indicate that it is not a good idea to do empathy style transfer on such backward dialog acts.

## 6 Conclusion

In this paper, we propose two novel few-shot prompting methods for empathy style transfer: target prompting and dialog-act-conditioned prompting. We examine and confirm the conditionality of empathy scores on the dialog acts of the source sentences. Target prompting is better at increasing empathy than prompting with traditional source/target pairs. Conditioning the few-shot prompt on the dialog act of the source sentence leads to stronger empathy transfer strength and better dialog act preservation. We expect that accounting for dialog acts will prove important for other style transfer tasks.

## Limitations

**Imbalanced and unrepresentative data**   There are two imbalances in the usage of the EC dataset: dialog act imbalance and sentiment imbalance. Despite using conversation-level mean empathy score as a selection criterion for test data, the distribution of dialog acts in the test set is still imbalanced (Figure 3b). The results in Table 4 may be biased towards the most frequent dialog acts, namely Statements, Questions, Forward Closings, and *none* texts. There are many other scenarios in the broader world where empathy style transfer is needed, such as medical and therapeutic conversations. Our results on tragic events are not representative of such scenarios, especially scenarios involving positive empathy. Additional data including more scenarios and positive empathy could expand on this work.

**Multiple dialog acts in one sentence**   Long texts tend to have multiple dialog acts because they often consist of multiple subsentences. In our analyses, we only use the dialog act label of the highest probability as the dialog act for each sentence. It is difficult to rewrite a sentence of multiple dialog acts while retaining its semantics. Though the sentences of multiple dialog acts only make up $4.5\%$ of the test samples, $12.0\%$ of the sentences in the EC dataset have multiple dialog acts. It is still debatable whether it is optimal to treat such sentences as single dialog acts and perform DA prompting in the same way as discussed in this paper, or whether it would be better to incorporate multiple dialog acts into few-shot examples instead.

**Ambiguous dialog acts**   Another bottleneck in conditioning style transfer on dialog acts is the existence of sentences with ambiguous dialog acts. Such sentences are classified with no dialog act label, and this might be more problematic than having multiple dialog acts since they make up a larger proportion of our data ($16.63\%$ in text data and $13.10\%$ in the EC dataset). Our current treatment of such sentences is to give them a *none* label and feed them the same few-shot examples used for non-DA methods, but this may not be optimal since DA methods improve performance.

## Ethical Considerations

Style transfer to increase empathy can help improve the quality of human and chatbot conversations in applications ranging from customer support to mental health. It can similarly potentially be used to increase the efficacy of manipulative behaviors.

It should also be noted that empathetic communication can be highly context-dependent. The work here ignores the many potentially important contexts: who is talking to whom about what. What is their relationship? Their identities? Their cultures? Their power relations? Communications that are perceived as empathetic in one context may be offensive or creepy in another. People also differ widely in their response to "empathetic" communications from chatbots in some contexts, with some people finding it offensive to have chatbots, rather than humans expressing empathy. Again, there are strong individual and cultural differences.

## Acknowledgements

We would like to thank the reviewers for their fruitful discussion with us. We would also like to thank Claire Daniele for editorial support.

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

## A Dialog acts used

The naming used in the explicit DA methods is shown in Table 5. The names were obtained by rewriting the DAMSL codes of Jurafsky et al. (1997) into their noun versions (See `https://web.stanford.edu/~jurafsky/ws97/manual.august1.html` for details).

| Dialog Act | Name |
|---|---|
| sd | non-opinion statement (fact) |
| sv | opinion statement (statement) |
| qh | rhetorical question |
| qo | open question |
| qr | or-question |
| qw | wh-question |
| qy | yes-no question |
| b | acknowledgement |
| ba | appreciation |
| bf | paraphrase |
| bh | rhetorical question continuer |
| br | request for repeat |
| by | expression of sympathy |
| fc | forward closing |
| fp | forward opening |
| ft | expression of gratitude |
| fx | explicit-performative |
| a | apology |
| aa | agreement |
| ad | action directive |
| na | affirmative answer |
| ng | negative answer |
| nn | no-answer |
| no | non-yes-no answer |
| h | evasive statement |
| % | interruption |
| + | sentence continuer |
| t3 | 3rd-party-talk |
| ^2 | collaborative completion |
| ^q | quotation |
| none | text |

Table 5: Full names of the dialog acts appeared in the EC dataset, obtained and modified from Jurafsky et al. (1997).

## B Empathy and Dialog Act Distribution

We present the empathy score distribution and dialog act histogram for the selected test samples in Figure 3. Though there are still dominant dialog acts in terms of proportion, the overall distribution is relatively less biased than taking the least empathetic sentences from the EC dataset, which mostly consists of Forward Closings according to Figure 2.

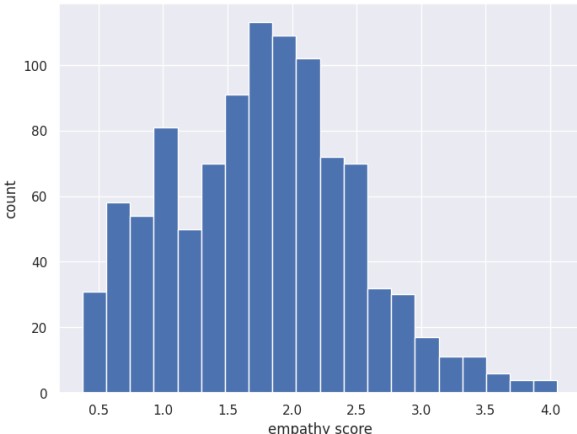

(a) Empathy score distribution for selected test samples

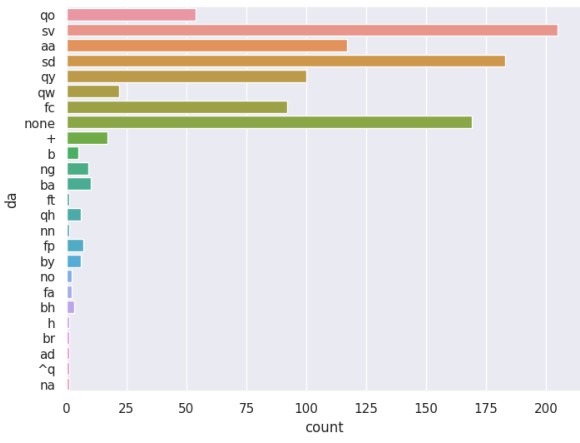

(b) Dialog act histogram for selected test samples

Figure 3: Empathy score distribution and dialog act histogram for the selected 1016 test samples. 63% of them have an empathy score below 2.0 (50% score threshold). A total of 25 dialog acts are included, with Opinions (*sv*), Facts (*sd*), and '*none*' taking the largest share of the test samples.

## C Empathy distributions for dialog acts

We present the histograms of empathy scores for the top 9 most frequently occurring dialog acts in the entire EC dataset in Figure 4. While statements and agreements have similar distributions that look like a normal distribution slightly skewed on the right, other dialog acts have very distinctive distributions. Some of the dialog acts (*by* and *qh*) have much smaller proportions in the EC dataset compared to the top 3 most frequently occurring ones, so further data collection of these dialog acts will be necessary to draw conclusions on their population distribution of empathy scores.

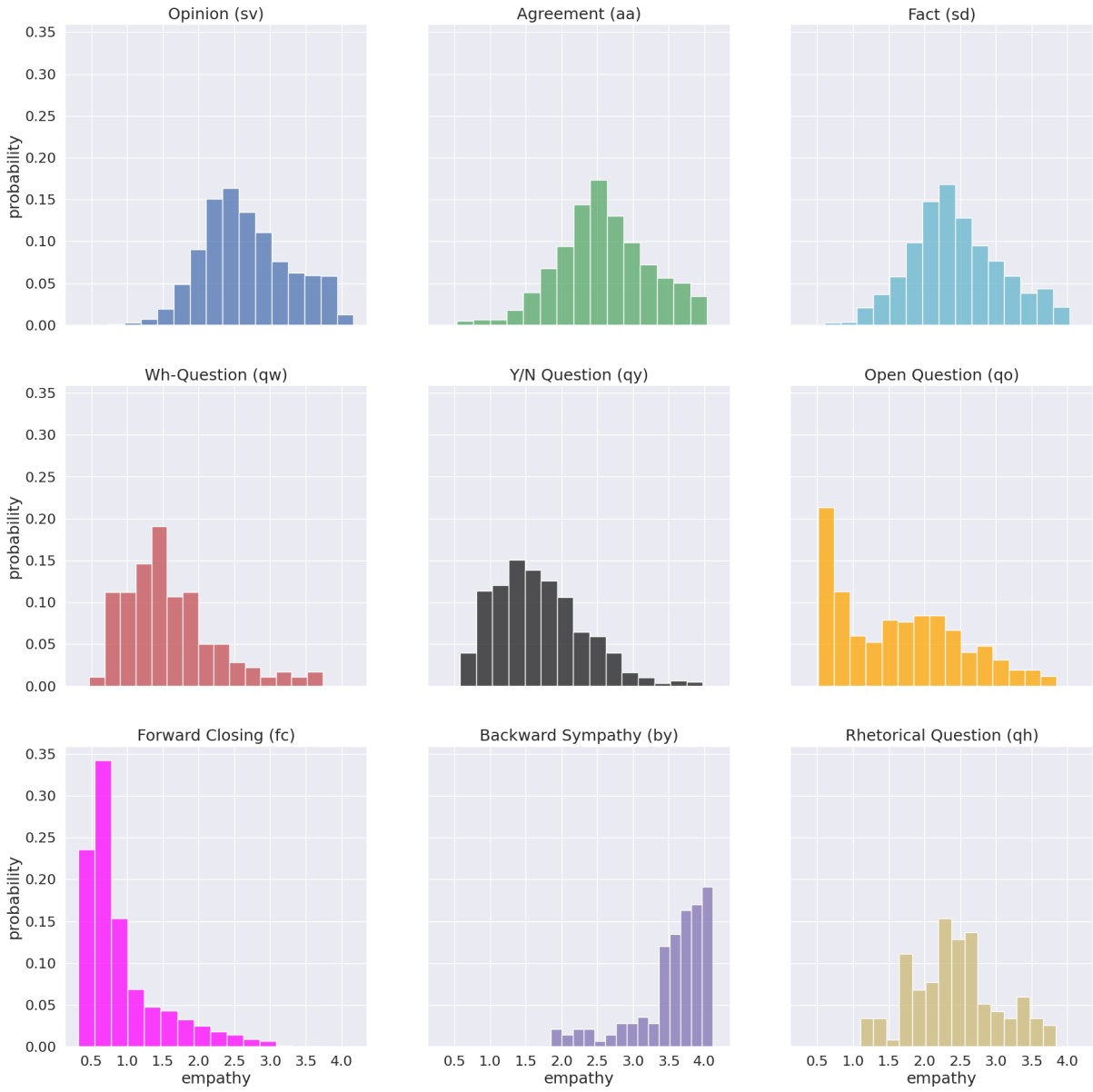

Figure 4: Histograms of empathy scores for the top 9 most frequent dialog acts in the EC dataset. These figures illustrate that Opinions, Facts, and Agreements share similar empathy score distributions while other different dialog acts have very different empathy score distributions.

## D Prompt formats and examples

We present the prompt formats and some samples of few-shot examples for Pairwise prompting and DA-Target prompting in Table 6, to highlight their differences.

## E DA-specific analyses

We present the results of all 7 prompting methods for each dialog act category in Figure 5, 6, and 7, See the main text for descriptions of these figures.

Figure 5 shows DA-specific evaluation results for ∆Emp, style transfer accuracy, and BLEURT.

Figures 6 and 7 show changes in dialog acts after empathy style transfer for forward communication and backward communication, respectively. Each subplot stands for a dialog act category of the source sentences mentioned in Table 3 where the Statement category is further separated into the Fact and Opinion categories because of their large shares in the dataset.

| Component | Format |
|---|---|
| Common Prefix | You are an excellent linguist. Your task is to rewrite the input {DA} in the most empathetic way possible. Make sure the input and output have the same content. Here are some examples: |
| Explicit Specification | You are an excellent linguist. Your task is to rewrite the input {DA} in the most empathetic way possible. Make sure the text is still a {DA} after rewriting. Make sure the input and output have the same content. Here are some examples of empathetic {DA} pairs (pairwise) / highly empathetic {DA} (target): |
| Few-shot examples (Pairwise) | Input: I think these children went through a bad time.
Output: My heart goes out to these innocent children who have endured such trying times, bearing the heavy weight of hardship and adversity.

Input: The Jewish people are experiencing something difficult.
Output: The Jewish community is navigating through deeply challenging circumstances, grappling with pain and emotional turmoil.

Input: Giving a child away is sometimes the right decision, though difficult. It can be hard, but it is the best choice for the child.
Output: It's an excruciatingly complex decision to entrust a child to another's care. Despite the heart-wrenching emotions, it's truly the most compassionate and selfless choice for the child's well-being and future. |
| Few-shot examples (DA-Target, Wh-Questions) | * What transpired in the lives of those precious, helpless infants that they had to undergo such experiences?

* What heart-wrenching circumstances might have led the family to take the agonizing decision to leave behind their beloved pets?

* How pervasive is this heartrending issue, in which these innocent children are struggling with hunger and fatigue, desperately trying to stay awake? |

Table 6: Prompt components. For direct prompting methods, the {DA} placeholder is replaced with "text". For DA-conditioned prompting methods, {DA} is replaced with the source dialog act name specified in Table 5.

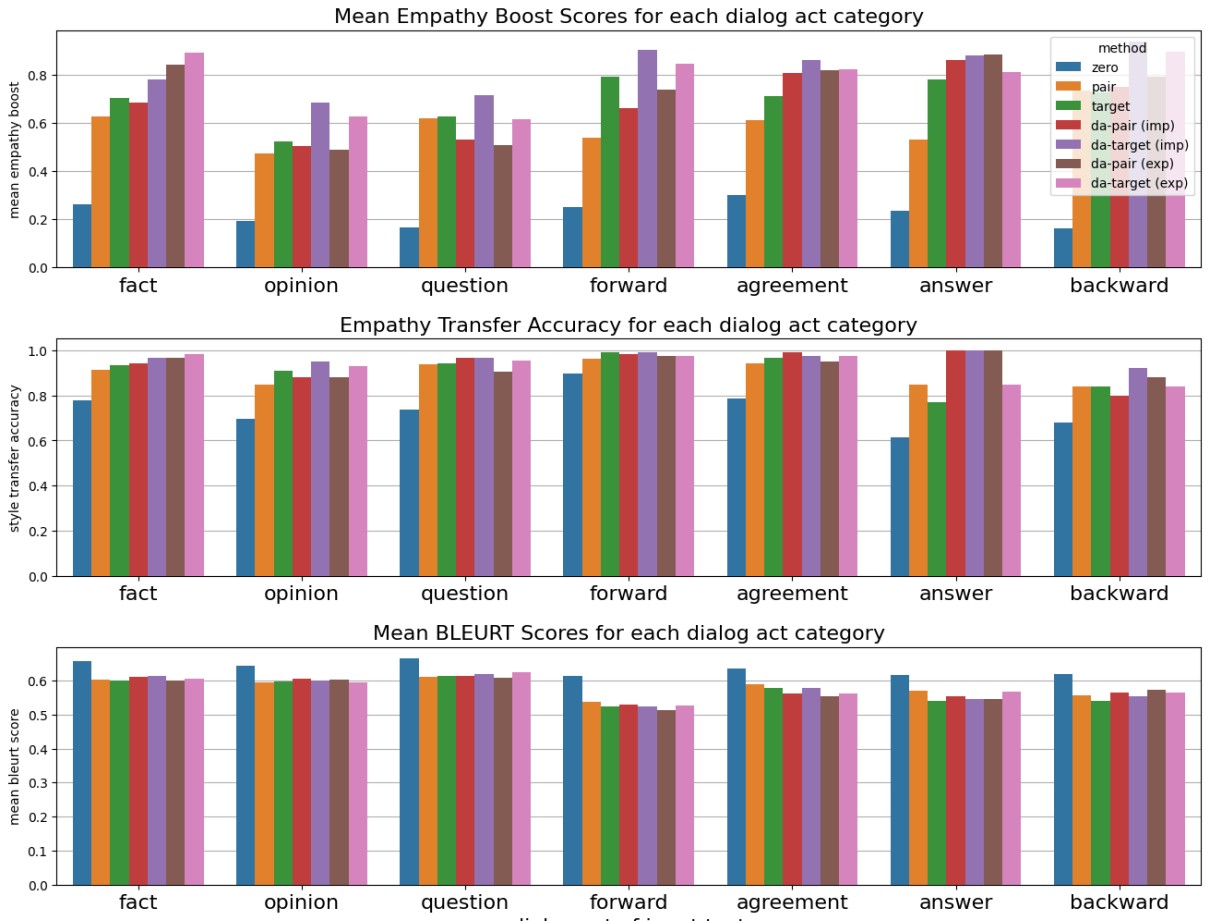

Figure 5: Barplots of different metrics for different methods on each dialog act category. Each subplot represents the results for one evaluation metric as suggested by the titles. The x-axis represents the 7 dialog act categories of the source sentences, excluding *none* category since its results do not contribute to the analysis. The y-axis represents the values on the metric. Each bar color stands for each prompting method. DA-Target prompting generally transfers better empathy at a higher accuracy compared to other methods, with the explicit version performing particularly well for facts and the implicit version for others. There is minimal difference in semantic similarity between the few-shot prompting methods.

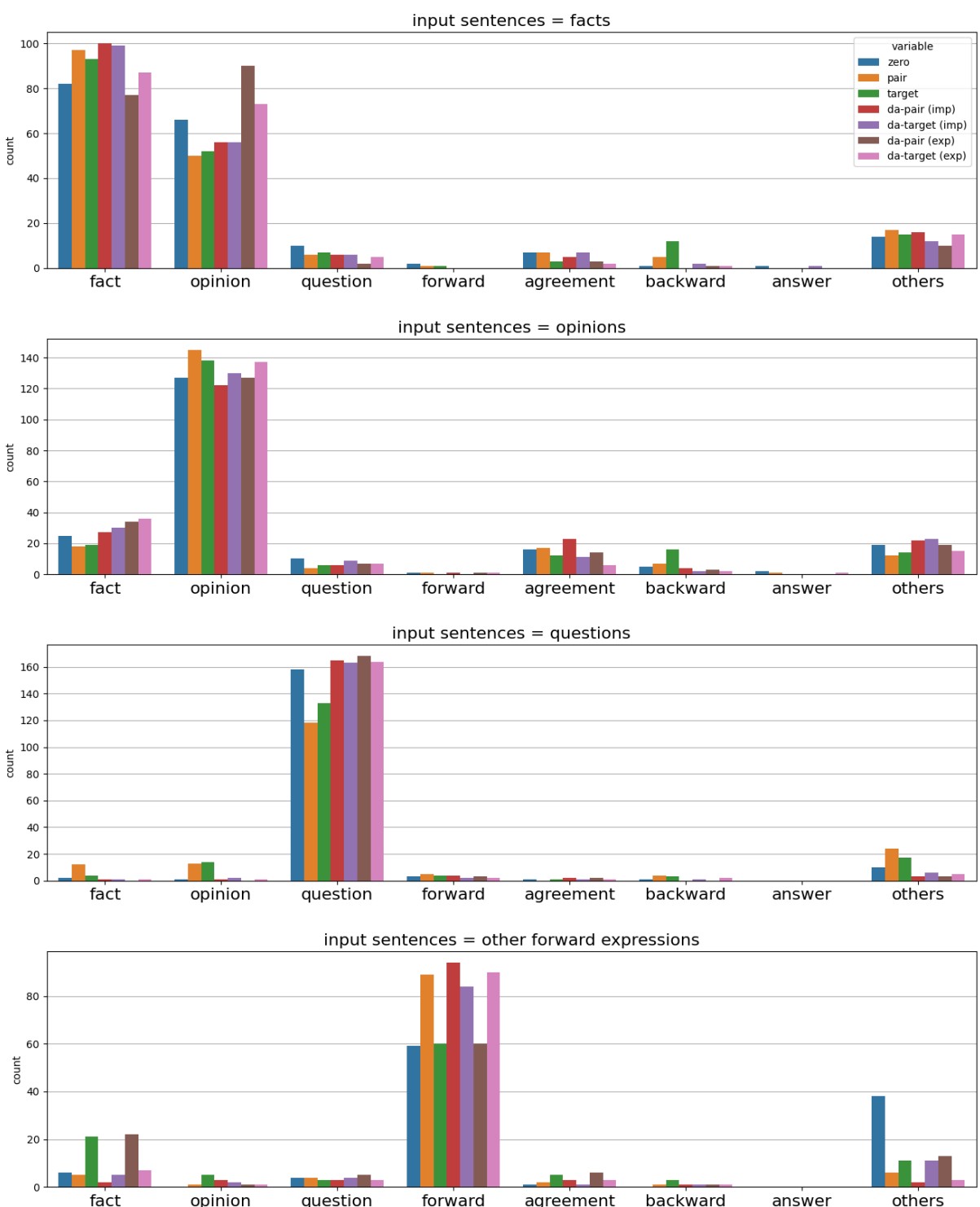

Figure 6: Countplots of dialog acts of generated sentences for forward source dialog act categories. Each subplot represents the count plot in the case where the input sentences are of the dialog act category specified in the title. The x-axis represents the 8 dialog act categories of the generated sentences (different from Figure 5), including *none* category since it contributes to the analysis of changes in dialog act. Each bar color stands for each prompting method. Facts are prone to be converted into opinions after empathy style transfer, while Questions and Other forward expressions are unlikely to be converted into different dialog act categories.

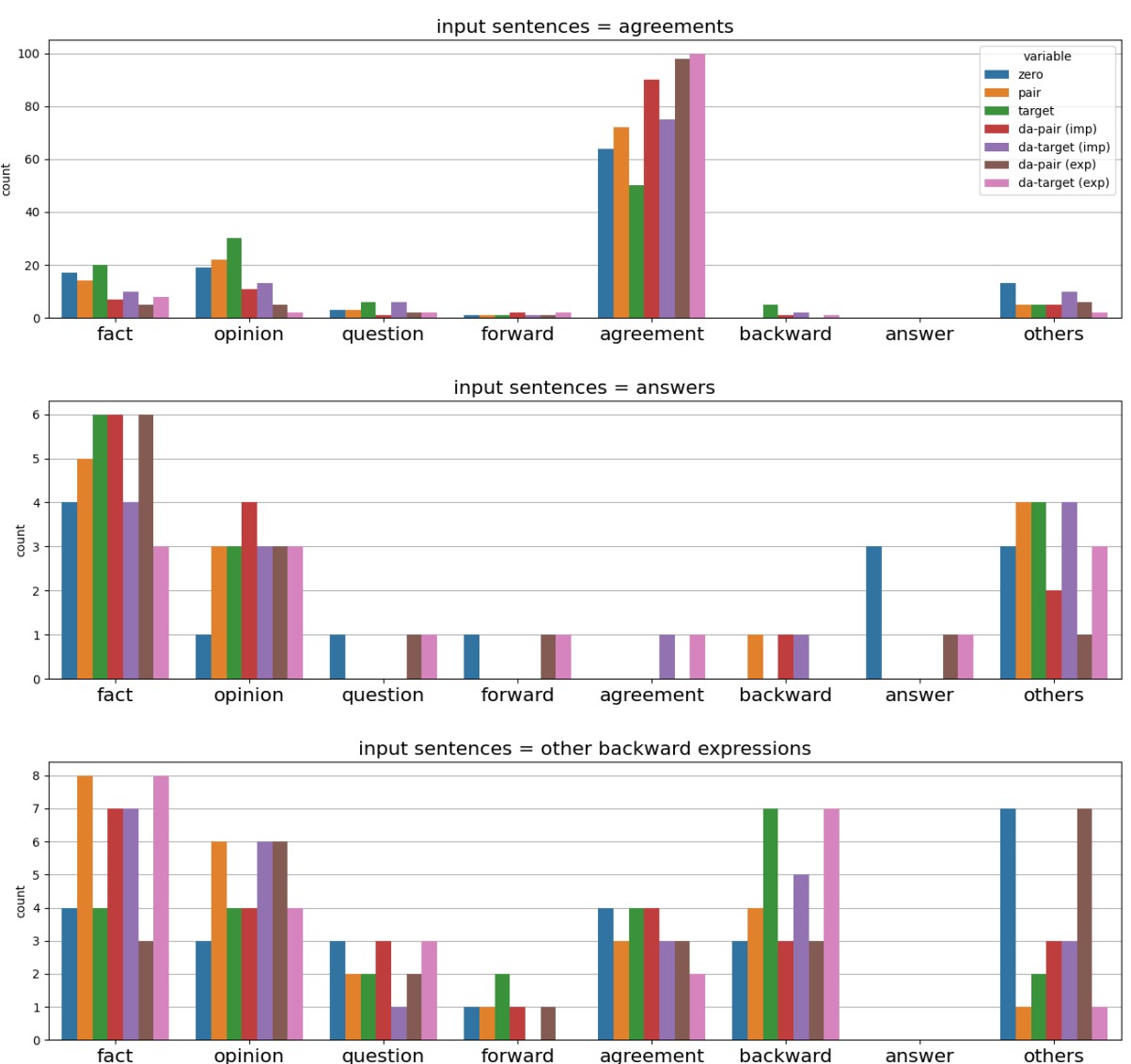

Figure 7: Countplots of dialog acts of generated sentences for backward source dialog act categories. The formats are the same as Figure 6. Unlike forward expressions, backward expressions are highly prone to changing after empathy transfer. Explicit methods show their advantage in keeping the dialog act of Agreements, but they did not perform well with Answers and Other backward expressions either.