# OpenReview forum: "Conditioning on Dialog Acts improves Empathy Style Transfer"
_EMNLP/2023/Conference — EMNLP 2023 Findings_

### Official Review · Reviewer_rbqx · 2023-08-04

**Soundness:** 4

**Excitement:**

3: Ambivalent: It has merits (e.g., it reports state-of-the-art results, the idea is nice), but there are key weaknesses (e.g., it describes incremental work), and it can significantly benefit from another round of revision. However, I won't object to accepting it if my co-reviewers champion it.

**Paper Topic And Main Contributions:**

In particular, the research examines the function of dialog actions in empathy style transfer, which entails altering a statement to make it more sympathetic while preserving its original meaning.

The major contribution of this study is the suggestion and assessment of dialog-act-conditioned prompting and target prompting as two innovative prompting techniques for empathy style transfer using GPT-4. While dialog-act-conditioned prompting guesses the dialog act of the source phrase and gives few-shot examples of the same dialog act to foster empathy, target prompting simply utilizes examples of the target style.

The study discovered that conditioning the prompt on dialog acts enhances the empathy transfer while better retaining source dialog acts, and target prompting creates stronger empathy than pairwise prompting while maintaining a similar degree of content. The study also emphasizes the significance of dialog actions in empathy style transmission and urges additional research into their function in style transfer.

**Questions For The Authors:**

1. Could you provide the theoretical support for the idea in sec 3.1?
2. How do you prove that the comparisons in the experiment are complete?
3. In the experiment section, is that possible to provide the variance of the experiments as well as the significance tests?

**Reasons To Accept:**

1. How to execute empathy style transfer via prompt-based inference is a critical research subject in natural language processing, which is addressed in this work.
2. Target prompting and dialog-act-conditioned prompting, two innovative prompting methodologies that the study suggests, may prove helpful for future NLP style transfer research.

**Reasons To Reject:**

1. The idea that the transfer between "apathetic" and "empathetic" is distinct from the transfer between "unempathetic" and "empathetic" with regard to the issue setting in sec. 3.1 may be based on intuition and prior understanding of empathy and related styles. Theoretical support for this supposition and an explanation of how empathy styles are arranged on a continuous, real-valued scale would be helpful, though. Without more support, this idea may be questioned and criticized.

2. While the comparisons of target vs. pairwise prompting and dialog-act-conditioned vs. direct prompting offer insights into the efficacy of various few-shot prompting strategies, it is unclear from these comparisons whether they represent a thorough analysis of all potential prompting methods or variations.

**Reproducibility:**

4: Could mostly reproduce the results, but there may be some variation because of sample variance or minor variations in their interpretation of the protocol or method.

**Reviewer Confidence:**

3: Pretty sure, but there's a chance I missed something. Although I have a good feel for this area in general, I did not carefully check the paper's details, e.g., the math, experimental design, or novelty.

---

> ### Author Rebuttal · Authors · 2023-08-29
>
> Thank you very much for your review. We reply to each of your concerns below.
>
> Reasons to Reject:
> 1. We believe this is a great point and will clarify this further in the revision. Many past papers on text style transfer assumed binary or categorical style labels (Toshevska and Gievska, 2021; Hu et al., 2022; Jin et al., 2022), and we attempted to explain that we assume numerical empathy style scores instead because empathy is too complicated to be discretized as discussed by the papers in line 695-697 and 731-734. We will delete line 201-203 and modify this section to reflect our assumption with more support.
>
> 2. In general, with prompting it is quite hard to show that we are able to have good coverage and conversely with open-source LLMs it is hard to claim they generalize to larger closed-source models. As a result, any paper (generally from academia) without access to large closed-source models has the drawback of being unable to show full prompt coverage. This having been said, we tried variations of zero/few-shot prompting together with more complex prompting methods (e.g., self-critique and chain-of-thought) during our initial experiments. We found that few-shot prompting provides a better representation of the text style transfer task because it clearly specifies how the input and output should look.
>
> Questions for the Authors:
> 1. We will clarify this further in the revision as discussed above.
>
> 2. We have 2 main branches of comparison: Target vs Pairwise, and DA-conditioned vs direct, and we also included Zero-shot prompting as our baseline. We assume by “complete” you mean the set of choices in each comparison is complete. Zero-shot prompting provides no input or output example, Target prompting provides only output examples, Pairwise prompting provides both input and output examples, and it makes no sense to use a method with only input examples, so the first comparison should be complete in choice. The second comparison is a binary comparison of whether to condition the examples on the input dialog act, thus complete in choice. The addition extension of “implicit vs explicit” is also a binary comparison of whether to specify the input dialog act label in the prompt, thus complete in choice. If you meant “completeness” in other aspects, we are happy to discuss further.
>
> 3. Yes. The t-test results (p-values) across all inputs are:
>
> - Target vs Pairwise comparison (5% significance level):
>     - Plain: 0.017
>     - DA-conditioned (implicit): 0.000 (3.856e-6)
>     - DA-conditioned (explicit): 0.016
>
> - DA-conditioned vs Plain comparison (1% significance level):
>     - Target: 0.000 (4.026e-5)
>     - Pairwise: 0.008
>
> - Implicit vs Explicit comparison (insignificant):
>     - Target: 0.434
>     - Pairwise: 0.142
>
> As shown above, our results for the two main contributions are statistically significant, while the extensions (of explicitly specifying dialog act labels) are not. The results align with our findings in line 454-458. We will run the significance tests specific to each dialog act category in the revision.

---

### Official Review · Reviewer_89A6 · 2023-08-05

**Typos Grammar Style And Presentation Improvements:** line 443： wed => we
**Soundness:** 3

**Excitement:**

4: Strong: This paper deepens the understanding of some phenomenon or lowers the barriers to an existing research direction.

**Paper Topic And Main Contributions:**

This paper investigates the prompting methods for empathy style transfer(incremental).  Based on GPT-4 model, It mainly proposed two few-shot methods for prompting: target prompting and dialog-act conditioned prompting.  The experiments are based on the Empathetic Conversation dataset and evaluated with three categories of metrics: Transfer Length, Content Preservation, and Fluency.  Extensive experiments and comparative study shows that target prompting is better than pairwise prompting on increasing empathy, and conditioning on the dialogue act of source sentence leads to stronger empathy and better dialogue act preservation.

**Questions For The Authors:**

1. Are you use three examples for all few-shot cases? Maybe I missed some content, but I saw there are three examples in the appendix Table 6.
2.  line 335-335, what do you mean by qualified examples for targeting prompting here? all highly empathic examples. Are you always using the same few-shot examples for all prompts for all inputs? Could you clarify more?
3. It makes sense that the qualified examples in the target prompting are from different dialogue acts. But i am curious what if you didn’t explicitly give the dialogue act, but only list a few examples all in target dialogue acts? What will happen? Do we still need the explicit dialogue act in that cases?
4. For each method, could please also clarify the effort used for prompting engineering?

**Reasons To Accept:**

It studies a novel direction on bringing dialogue acts for empathy style transfer. The proposed targeting prompting and conditioning on the dialogue act of the source sentence works great. The overall evaluation is extensive.

**Reasons To Reject:**

It is unclear what are the qualified few-shot examples selected for target prompting(see more details in the questions).  It seems many factors here impact the performance: few-shot example selection(number of examples, diversity, distribution of empathy scores), prompt templates etc. The discussion to control those factors is not well-organized,  it is not convincingly clear that claim the proposed prompting method is better than others.

**Reproducibility:**

3: Could reproduce the results with some difficulty. The settings of parameters are underspecified or subjectively determined; the training/evaluation data are not widely available.

**Reviewer Confidence:**

4: Quite sure. I tried to check the important points carefully. It's unlikely, though conceivable, that I missed something that should affect my ratings.

---

> ### Author Rebuttal · Authors · 2023-08-29
>
> Thank you very much for your review. We reply to each of your concerns below.
>
> Reasons to Reject:
> - While it is difficult to include all examples in the paper because of the number of dialog acts involved, we will add more to the appendix in the revision. We apologize for the confusion in describing our experimental controls, and we will clarify this next revision.
>
> Questions for the Authors:
> 1. Yes. We apologize for the confusion. We will clarify this.
> 2. For plain target prompting, the qualified examples (i.e., highly empathetic examples) are always the same for all inputs. However, for DA-Target prompting, the qualified examples are only the same for the same source dialog act (e.g., if two inputs are both opinions, they share the same examples. If one is an opinion and the other is a question, they use different examples.) We will elaborate further in the revision.
> 3. This is the “implicit” method, and the “explicit” specification of dialog act labels is an extension of our DA-conditioned methods (line 231-249). No conclusion could be drawn from the comparison “implicit vs explicit” (line 454-474), meaning that dialog act label specification does not necessarily improve the performance when we condition the few-shot examples on the dialog act of the input sentence. The significance test results in the rebuttal for Reviewer rbqx indicate this as well.
> 4. We clarified the effort used for prompt engineering in line 308-351 and Table 6. We will attempt to improve the clarity in the revision.
>
> Typos Grammar Style And Presentation Improvements:
> - We apologize for the typos and will fix them.

---

### Official Review · Reviewer_kZET · 2023-08-11

**Soundness:** 2

**Excitement:**

2: Mediocre: This paper makes marginal contributions (vs non-contemporaneous work), so I would rather not see it in the conference.

**Paper Topic And Main Contributions:**

This paper attempts to make utterances more empathetic by prompting GPT-4 with few-shot samples and dialogue act labels. The authors show that it is unnecessary to provide source-target pairs when prompting GPT-4 and dialogue act labels might be a help.

**Questions For The Authors:**

* How about comparing the performance of giving emotion information vs. dialogue acts? I suspect if you give any sorts of related info with the source sentence or context can lead to performance improvements.
* It would be more informative to clearly flesh out what the authors want to highlight as their main contribution. Simply saying “Dialogue acts matters” is ambiguous and this information isn't something that readers in the NLP field will find novel. What is the most important finding of this paper?

**Reasons To Accept:**

Rephrasing sentences to have more empathy can impact real-world applications.

**Reasons To Reject:**

* Empathy is very difficult to be captured by automatic metrics, hence human evaluation is a must to verify the improvements. However, the authors only report automatic metrics. Moreover, the difference of the automatic scores between approaches are small. Therefore it is hard to tell whether the approach in the paper is effective or not.

* Although the authors highlight dialogue act labels to be their main contribution, the results do not favor their claim, because the scores are better when the dialogue act label for the source target is not given (implicit vs. explicit). Moreover, the plain Target prompting, which do not include dialogue act labels, is comparable to DA-Pairwise. Again, further human evaluation is needed to compare the true effectiveness of the authors’ approach.

**Reproducibility:**

4: Could mostly reproduce the results, but there may be some variation because of sample variance or minor variations in their interpretation of the protocol or method.

**Reviewer Confidence:**

3: Pretty sure, but there's a chance I missed something. Although I have a good feel for this area in general, I did not carefully check the paper's details, e.g., the math, experimental design, or novelty.

**Typos Grammar Style And Presentation Improvements:**

- Some sentences were hard to follow. Why do you need line 198?
- Many typos (e.g., wed use, capitalization errors) and citation formatting errors in sentences.
- It would be much better to clearly indicate which approach you are referring to in the results section (e.g., explicitly mention the method name in the table).

---

> ### Author Rebuttal · Authors · 2023-08-29
>
> Thank you very much for your review. We reply to each of your concerns below.
>
> Reasons to Reject:
> - We agree that human evaluation is important; however, Omitaomu et al. (2022) showed that there is a correlation of 0.771 using 5-fold cross-validation between the empathy evaluator and human judgments. This was further validated in the WASSA 2023 shared task. While the difference in semantic similarity scores is small due to the high performance of GPT-4, this is not the case for transfer strength. Figure 5 in the Appendix shows a clearer distinction in empathy score boost between different prompting methods on different source dialog acts. While the difference in accuracy may look small, we scaled the y-axis of the graph to the range [0,1] instead of using our baseline (zero-shot GPT-4, the blue bar) as the reference point, which occupies 75.98% of the y-axis on average as stated in Table 4. We will fix this when we resubmit.
> - Our main contribution is to highlight the importance of "dialog act"  in empathy style transfer (line 106-110, 231-242, 580-583), instead of "dialog act labels”. This is the comparison “DA-conditioned vs Direct” (line 442-453), meaning whether or not the few-shot examples in the prompt have the same dialog act as the source sentence. We will clarify this in future iterations of the paper.
> - We stated that the comparison “Implicit vs Explicit" is not our main contribution because no clear conclusion can be drawn from this (line 243, 454-458). The significance test results in the rebuttal for Reviewer rbqx indicate this as well.
> - We believe the comparison between Target and DA-Pairwise prompting is unrealistic because it messes up the control of variables when comparing methods in experiments. We have 2 branches of comparison: Target vs Pairwise, and DA-conditioned vs plain. When the second branch is fixed, we find that target methods transfer empathy better than pairwise methods. That is our contribution.
>
> Questions for the Authors:
> - While other related information may lead to performance improvements, this is beyond the scope of our research. As indicated in Section 2, our research links the following 3 aspects of NLP altogether, which no published paper has done to our knowledge:
>     - Empathy: a major emotional or cognitive response in conversations
>     - Text style transfer: a challenging subfield of natural language generation, unsolved by zero-shot prompting with LLMs
>     - Dialog act: a key factor of a sentence's functionality in a given conversation
>
>     We are more than excited to explore other related information that helps with empathy transfer in the future, such as personality traits, other emotions, and other sentence attributes as well. Nonetheless, this is outside the scope of our paper.
>
> - Our main contributions are fleshed out in line 103-110, 213-242, 285-307, 426-453, and 573-585. While the other results in Section 5 and Appendix may not appear related to our main contributions on the surface, they are closely aligned with our main contribution (line 528-535, Figure 5-7). In line 106-110, we stated that conditioning the prompt (input to GPT-4) on the dialog act of the source sentence would improve GPT-4’s performance in empathy style transfer. This is what we mean by “Dialog acts matter”.
>
> Typos Grammar Style And Presentation Improvements:
> - Many past papers on text style transfer assumed binary or categorical style labels (Toshevska and Gievska, 2021; Hu et al., 2022; Jin et al., 2022). In line 198 we attempted to explain that we assume numerical empathy style scores instead. We will clarify this in our revision.
> - Thank you for pointing out the typos we will fix them.
> - We have clearly indicated the method names in Table 2 and 4.

---

### Meta-Review · Area_Chair_CCWf · 2023-09-19

**Recommendation:** 4

**Metareview:**

This paper attempts to make utterances more empathetic by prompting GPT-4 with few-shot samples and dialogue act labels. The authors show that it is unnecessary to provide source-target pairs when prompting GPT-4 and dialogue act labels might be a help.

pros:
- It studies a novel direction on bringing dialogue acts for empathy style transfer. The proposed targeting prompting and conditioning on the dialogue act of the source sentence works great. The overall evaluation is extensive.

cons:
- Empathy is very difficult to be captured by automatic metrics, hence human evaluation is a must to verify the improvements. However, the authors only report automatic metrics. Moreover, the difference of the automatic scores between approaches are small. Therefore it is hard to tell whether the approach in the paper is effective or not.
- It is unclear what are the qualified few-shot examples selected for target prompting(see more details in the questions). It seems many factors here impact the performance: few-shot example selection(number of examples, diversity, distribution of empathy scores), prompt templates etc. The discussion to control those factors is not well-organized, it is not convincingly clear that claim the proposed prompting method is better than others.
-  The idea that the transfer between "apathetic" and "empathetic" is distinct from the transfer between "unempathetic" and "empathetic" with regard to the issue setting in sec. 3.1 may be based on intuition and prior understanding of empathy and related styles. Theoretical support for this supposition and an explanation of how empathy styles are arranged on a continuous, real-valued scale would be helpful, though. Without more support, this idea may be questioned and criticized.

---

### Decision · Program_Chairs · 2023-10-07

**Decision:**

Accept-Findings

**Comment:**

This paper attempts to make utterances more empathetic by prompting GPT-4 with few-shot samples and dialogue act labels. The authors show that it is unnecessary to provide source-target pairs when prompting GPT-4 and dialogue act labels might be a help.

pros:
- It studies a novel direction on bringing dialogue acts for empathy style transfer. The proposed targeting prompting and conditioning on the dialogue act of the source sentence works great. The overall evaluation is extensive.

cons:
- Empathy is very difficult to be captured by automatic metrics, hence human evaluation is a must to verify the improvements. However, the authors only report automatic metrics. Moreover, the difference of the automatic scores between approaches are small. Therefore it is hard to tell whether the approach in the paper is effective or not.
- It is unclear what are the qualified few-shot examples selected for target prompting(see more details in the questions). It seems many factors here impact the performance: few-shot example selection(number of examples, diversity, distribution of empathy scores), prompt templates etc. The discussion to control those factors is not well-organized, it is not convincingly clear that claim the proposed prompting method is better than others.
-  The idea that the transfer between "apathetic" and "empathetic" is distinct from the transfer between "unempathetic" and "empathetic" with regard to the issue setting in sec. 3.1 may be based on intuition and prior understanding of empathy and related styles. Theoretical support for this supposition and an explanation of how empathy styles are arranged on a continuous, real-valued scale would be helpful, though. Without more support, this idea may be questioned and criticized.